# Selenide-catalyzed enantioselective synthesis of trifluoromethylthiolated tetrahydronaphthalenes by merging desymmetrization and trifluoromethylthiolation

Jie Luo[1], Qingxiang Cao[1], Xiaohui Cao[1] & Xiaodan Zhao [1]

Trifluoromethylthiolated molecules are an important class of biologically active compounds and potential drug candidates. Because of the lack of efficient synthetic methods, catalytic enantioselective construction of these molecules is rare and remains a challenge. To expand this field, we herein disclose a bifunctional selenide-catalyzed approach for the synthesis of various chiral trifluoromethylthiolated tetrahydronaphthalenes bearing an all-carbon quaternary stereocenter with gem-diaryl-tethered alkenes and alkynes by merging desymmetrization and trifluoromethylthiolation strategy. The products are obtained in high yields with excellent enantio- and diastereo-selectivities. This method can be applied to the desymmetrization and sulfenylation of diols as well. Computational studies reveal that selenide can activate the electrophilic reagent better than sulfide, confirming the higher efficiency of selenide catalysis in these reactions. On the basis of the theoretical calculations, an acid-derived anion-binding interaction is suggested to exist in the whole pathway and accounts for the observed high selectivities.

[1] Institute of Organic Chemistry and MOE Key Laboratory of Bioinorganic and Synthetic Chemistry, School of Chemistry, Sun Yat-Sen University, Guangzhou 510275, PR China. Correspondence and requests for materials should be addressed to X.C. (email: caoxh5@mail.sysu.edu.cn) or to X.Z. (email: zhaoxd3@mail.sysu.edu.cn)

In recent years, many efforts have been devoted to the incorporation of fluorine atoms or fluorine-containing groups such as trifluoromethyl ($CF_3$), trifluoromethoxy ($CF_3O$), and trifluoromethanesulfenyl ($CF_3S$) ones into the parent molecules for various purposes because of the fluorine effect[1–6]. Among these endeavors, strategic synthesis of $CF_3S$ molecules has been paid special attention owing to the strong electron-withdrawing effect and extremely high lipophilicity value ($\pi_R = 1.44$) of $CF_3S$ group[5–12]. However, little success has been achieved on enantioselective trifluoromethylthiolation until now, although stereogenic $CF_3S$ molecules warrant further studies considering the importance of chiral centers in medicine[13–20]. Thus, developing new methods to create versatile chiral $CF_3S$ molecules, especially those with an all-carbon quaternary stereocenter through a novel and enantioselective reaction mode, is highly desirable.

Catalytic enantioselective desymmetrization is an attractive strategy for the construction of chiral all-carbon quaternary stereocenters by the conversion of prochiral quaternary carbon centers[21–23]. Using this strategy, numerous valuable, potentially bioactive molecules having a chiral all-carbon quaternary center can be quickly accessed from different functionlized starting materials[24–43]. In particular, olefinic or alkynyl carboxylic acids[33,34], alcohols[35–39], and amines[40–43] were frequently employed as the substrates to undergo enantioselective desymmetrization and cyclization to generate heterocycles by metal- or organocatalysis (Fig. 1a). In these transformations, the tethered nucleophile played an important role that it could bind a catalyst to guarantee an effective attack toward the multiple bond, which led to the formation of chiral products with high enantioselectivities. In contrast, enantioselective desymmetrization involving the attack of aryl group toward a multiple bond that results in the formation of multisubstituted tetrahydronaphthalene derivatives, an important class of bioactive compounds[44–46], has been far less explored possibly because of the lack of the appropriate interaction between the aryl moiety and catalyst[47–49]. Only a few relevant examples have been reported by Chemler who utilized amine- or hydroxy-tethered alkenes for carboamination and etherification through a copper-catalyzed radical pathway (Fig. 1b)[50–53].

Continuing our interest in Lewis basic selenium[54–62]-catalyzed trifluoromethylthiolation[19,20,63–65], we intended to produce chiral $CF_3S$ molecules with an all-carbon quaternary stereocenter through an enantioselective, electrophilic desymmetrization, and trifluoromethylthiolation mode. We envisioned that when gem-diaryl-tethered alkenes were employed as the substrates, the aryl group on substrate could act as a nucleophile to attack chiral selenide-captured trifluoromethylthiiranium moiety to directly afford chiral $CF_3$ tetrahydronaphthalenes (Fig. 1c). To cope with the main difficulty in this transformation, a proper chiral catalyst is essential that can control the attacking environment of the aryl group and thus induce the enantioselectivity of multiprochiral centers. Herein, we report our effort that gem-diaryl-tethered alkenes can undergo enantioselective desymmetrization and difunctionalization to efficiently afford $CF_3S$-tetrahydronaphthalene derivatives with bifunctional selenide catalyst. The generated products contain one chiral quaternary carbon center and other two stereocenters. The developed method can be applied to enantioselective desymmetrization and sulfenylation of diols as well.

## Results

**Initial Attempts and Optimization of Reaction Conditions**. We began our study of the electrophilic desymmetrization with 2,2-diphenyl olefinic benzamide **1a** as the model substrate. It could be easily synthesized from diphenylacetonitrile, and possesses two phenyl groups as a nucleophile and an extra benzamide group. To test the desymmetrization of **1a**, highly reactive electrophilic $(PhSO_2)_2NSCF_3$ as the $CF_3S$ source and bifunctional catalyst **C1** based on indane scaffold were utilized (Table 1). Based on our former observations[20], selenide **C1** with a triflic amide group was quite efficient for the trifluoromethylthiolation with the aid of acid. Pleasingly, at room temperature, the corresponding product **2a** was smoothly formed rather than amination product from benzamide group in 94% nuclear magnetic resonance (NMR) yield with 89% ee and 5:1 dr using trimethylsilyl trifluoromethylsulfonate (TMSOTf) as the acid. Lowering the reaction temperature to −78 °C could quickly improve the enantioselectivity to 97% ee with unchanged diastereoselectivity (Table 1, entry 2). It is noted that sulfide catalyst **C2** was not effective for this transformation at all under the similar conditions (Table 1, entry 3). To improve the diastereoselectivity of **2a**, various aryl selenides based on **C1** were tested for the reaction. While para-substituted phenyl group and meta-substituted phenyl group on the selenide had little influence, ortho-substituent on the phenyl ring largely enhanced the selectivity (Table 1, entries 5–8). To our delight, catalyst **C7** bearing both ortho-methyl and methoxy groups was highly efficient to afford **2a** in 99% yield with 99% ee and 50:1 dr. Using the mixed solvents of $CH_2Cl_2$ and $(CH_2Cl)_2$, the enantioselectivity of product **2a** could be improved to > 99% (Table 1, entry 9). In addition, other acids including both Lewis acid or BrØnsted acid gave slightly lower enantioselectivity (Table 1, entries 10–12). It is noteworthy that the reaction could not go to completion and the corresponding product was formed in moderate selectivity under the optimal conditions when the substrate derived from **1a** by further

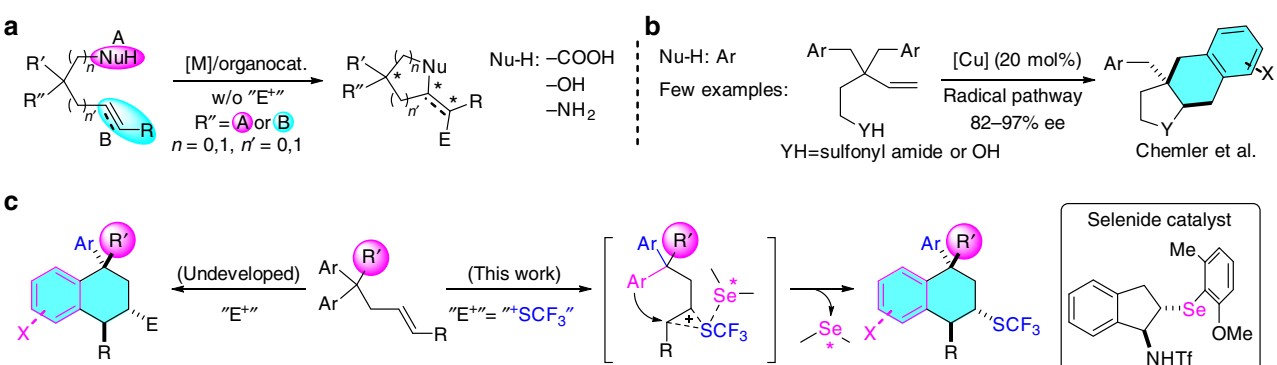

**Fig. 1** Enantioselective construction of all-carbon quaternary center-containing molecules via desymmetrization. **a** Known strategies for enantioselective desymmetrization. **b** Desymmetrization through copper-catalyzed radical pathway. **c** Enantioselective desymmetrization and trifluoromethylthiolation using aryl group as a nucleophile

**Table 1 Screening of reaction conditions**

| Entry | T (°C) | Cat. | Yields | ee | dr |
|---|---|---|---|---|---|
| 1 | rt | C1 | 94 | 89 | 5:1 |
| 2 | -78 | C1 | >99 | 97 | 5:1 |
| 3 | -78 | C2 | All starting material remains. | | |
| 4 | -78 | C3 | >99 | 97 | 5:1 |
| 5 | -78 | C4 | 93 | 98 | 13:1 |
| 6 | -78 | C5 | 88 | 98 | 38:1 |
| 7 | -78 | C6 | 57 | >99 | 18:1 |
| 8 | -78 | C7 | >99 | 99 | 50:1 |
| 9[*] | -78 | C7 | >99 | >99 | 50:1 |
| 10[*,†] | -78 | C7 | >99 | 92 | 33:1 |
| 11[*,‡] | -78 | C7 | >99 | 98 | 50:1 |
| 12[*,§] | -78 | C7 | 80 | 99 | 50:1 |

C1  C2 | C3  C4  C5  C6  C7

Bz C$_6$H$_5$CO, Tf CF$_3$SO$_2$, TMSOTf Me$_3$SiOSO$_2$CF$_3$, HPLC high-performance liquid chromatography, NMR nuclear magnetic resonance. Conditions: 1a (0.05 mmol), (PhSO$_2$)$_2$NSCF$_3$ (1.5 equiv), catalyst (20 mol%), TMSOTf (1.0 equiv), CH$_2$Cl$_2$ (2.0 ml), 12 h. Yield refers to NMR yield using trifluoromethylbenzene as the internal standard. The ee value was determined by HPLC analysis on a chiral stationary phase. The dr value was determined by crude $^{19}$F NMR. *Mixed solvents of 1 ml CH$_2$Cl$_2$ and 1 ml (CH$_2$Cl)$_2$ were used. †BF$_3$·OEt$_2$ (1.0 equiv) as the acid. ‡TfOH (1.0 equiv) as the acid. §Tf$_2$NH (1.0 equiv) as the acid

protecting nitrogen with methyl group was used (63% ee, see Supplementary Table 3 for details).

**Desymmetrization and Trifluoromethylthiolation**. With the optimal conditions in hand, we began to explore the substrate scope (Table 2). To ensure the full consumption of starting materials, 20 mol% of the catalyst loading was utilized for the transformations. Various aryl substituted olefins were first tested. All of them gave the corresponding products in good to excellent yields with excellent enantio- and diastereoselectivities (2a–h, 74–99% yields, 98–99% ees). Moreover, modified conditions were required for some substrates to give better yields or slightly better enantioselectivities. For example, the reactions could not go to completion under the optimal conditions for the formation of 2b–2d most likely because the weakly electron-withdrawing aryl group on the double bond eroded its reactivity toward CF$_3$S cation. When the reaction temperature was raised to −60 °C, all these substrates were fully converted. Besides, low catalyst loading (10 mol%) and low concentration were appropriate for the generation of 2e and 2h to suppress the possible attack of the electron-rich aryl group of catalyst toward the iranium ion. It was worthy to mention that a substrate bearing ortho-methyl-substituted phenyl group still gave the desired product in excellent yield with excellent enantioselectivity in spite of the steric hindrance around the double bond (2f, 94% yield, >99% ee). Enantioselective desymmetrizaiton of alkyl-substituted olefins was carried out under the similar conditions. Substrates bearing methyl or phenylethyl group gave the corresponding products in good yields with excellent ees (2i, 97% ee; 2j, 97% ee). To our surprise, gem-dialkyl-substituted olefins could efficiently afford the products bearing another achiral quaternary carbon center with excellent enantioselectivities (2k, 92% ee; 2l, 97% ee), although large steric hindrance might affect the cyclization.

Moreover, the developed method was also suitable for alkyne-derived compounds. Olefinic products were obtained in good yields. When phenyl-substituted substrate was utilized in the reaction, product 2m was formed with excellent ee (95% ee). The ethyl-substituted substrate gave 2n with a little lower ee (87%). These products contain a double bond, which can provide an opportunity for their further transformations. The absolute configuration of products was assigned to be 1R, 3S, 4S based on the X-ray crystallographic study of 2a.

The effect of functional groups attached to the quaternary carbon center on substrates was investigated (Table 2). When substrate 1o with more acidic proton was used, the reaction proceeded efficiently to afford the carbocyclization product 2o. In contrast, when the nitrogen of 1o was protected by methyl group, the corresponding substrate 1o′ gave product 2o′ with lower enantioselectivity (85% ee). It was noted that when the phenyl group attached to the double bond on 1o was replaced by an alkyl group, CF$_3$S-amination product was observed along with the formation of carbocyclization product. Free hydroxyl group on substrate had an impact on the enantioselectivity (2p, 81% ee). Compared to the reaction of 4-nitro-benzenesulfonamide (NsNH)-functionalized substrate, the decrease of enantioselectivity might attribute to OH-induced inappropriate H-bonding interaction between substrate and catalyst. When the hydroxyl group was protected by benzoyl or acyl group, the cyclization proceeded efficiently to produce the products with excellent ees (2q–s, 94–97% ees). It was noteworthy that the reaction of 1q was incomplete and afforded product 2q with 96% ee at −78 °C. Unexpectedly, when R′ group was hydrogen, the desired product 2t was still generated in 81% yield with 86% ee.

We then turned our attention to the desymmetrization with different gem-diaryl-tethered alkenes. Substrates with para- or ortho-substituted phenyl group at the quaternary carbon center gave the products in high yields with >99% ees under the similar

**Table 2 Enantioselective desymmetrization and trifluoromethylthiolation of *gem*-diaryl tethered alkenes/alkynes**

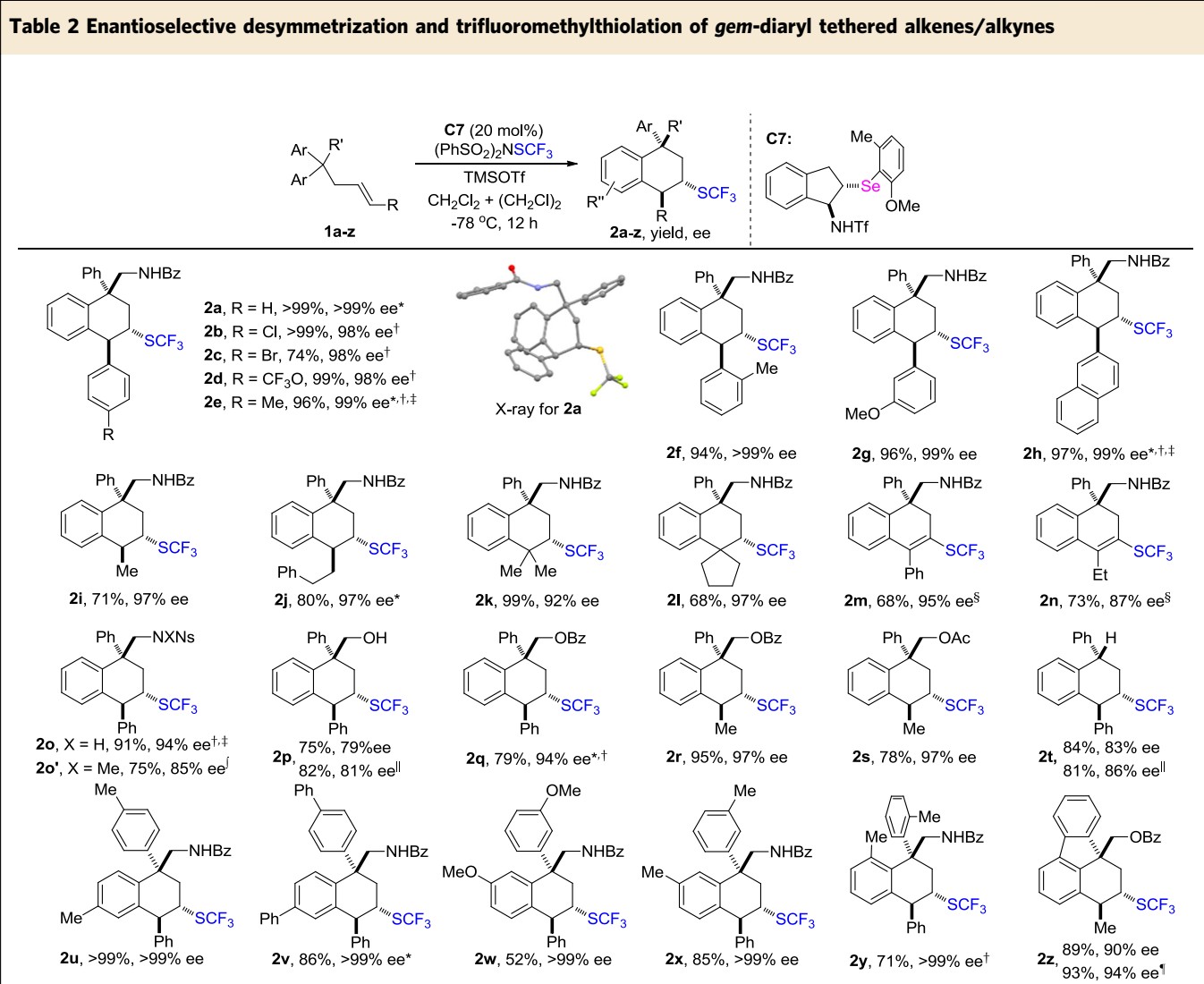

Bz C$_6$H$_5$CO, Ns 4-NO$_2$C$_6$H$_4$SO$_2$, Ac, CH$_3$CO, Tf CF$_3$SO$_2$, TMSOTf Me$_3$SiOSO$_2$CF$_3$, TIPSOTf $^i$Pr$_3$SiOSO$_2$CF$_3$. Conditions: 1 (0.10 mmol), (PhSO$_2$)$_2$NSCF$_3$ (1.5 equiv), TMSOTf (1.0 equiv), CH$_2$Cl$_2$ (2.0 ml) + (CH$_2$Cl)$_2$ (2.0 mL), −78 °C, 12 h. Yield is isolated yield. Ratio of ee was determined by HPLC analysis on a chiral stationary phase. Ratio of dr was determined by crude $^{19}$F NMR. Without note, diastereoselectivity is >99:1. *With 50:1 diastereoselectivity. †Reaction temperature: −60 °C. ‡CH$_2$Cl$_2$ (4.0 ml) + (CH$_2$Cl)$_2$ (4.0 ml) as the solvent; 10 mol% catalyst was used. §TMSOTf (2.0 equiv) was added. ⫽With 8:1 diastereoselectivity. ‖TIPSOTf (1.0 equiv) instead of TMSOTf. ¶BF$_3$·OEt$_2$ (2.0 equiv) instead of TMSOTf

conditions (**2u**, **2v**, and **2y**). When substrates with *meta*-substituted phenyl group at the quaternary carbon center were utilized, regioisomeric products were formed because of the site selectivity. The major isomer could be isolated with extremely high ees (**2w**, >99% ee; **2x**, >99% ee). Fluorene-derived alkene underwent desymmetrization and cyclization to generate product **2z** efficiently as well.

**Practicability of the Developed System**. To test the generality of the developed method, alkene **3** with more flexible benzyl groups was examined under the similar conditions (Fig. 2a). Product **4** was formed in high yield with good enantioselectivity. When this method was applied to the desymmetrization and sulfenylation of **1a** with sulfenylating reagents, no reaction occurred. This result was unexpected since the carbosulfenylation of alkenes has been realized by chiral selenophosphoramide catalysis[66–68]. Moreover, when olefinic diols were treated with sulfenylating reagent **6** in the presence of catalyst **C7**, thioproduct **7** was obtained in 67% with 92% ee and 9:1 *dr* via desymmetrization (Fig. 2b). The result shows that the developed reaction system has great potential for

electrophilic functionalization of alkenes with different electrophilic reagents, and thus will trigger more explorations using the similar conditions.

To further test the practical utility of the method, the reaction was scaled up with low catalyst loading. For example, desymmetrization of **1a** (1.0 g) afforded product **2a** (1.23 g) in 99% yield with excellent enantioselectivity (>99% ee) using 2 mol% **C7** (Fig. 2c). This desymmetrization reaction could run at the room temperature, and was rapidly completed within 5 min using catalyst **C7** to give product without much erosion of the selecticity. This result enhances the practicability of the method out of the lab. The recycle of the catalyst was also investigated. Alkene **1r** was chosen as the substrate because of its easy separation from the catalyst (Fig. 2d). During the recycling, the product was obtained in high yield for each time, and its enantioselectivity remained unchanged. After being recycled five times, 92% catalyst was still recovered.

The functional groups on substrates not only helped to enhance the selectivity of the reaction, but also offered us a great opportunity to pursue further transformations of products. Some synthetic applications of **2a** are depicted in Fig. 3 and all the

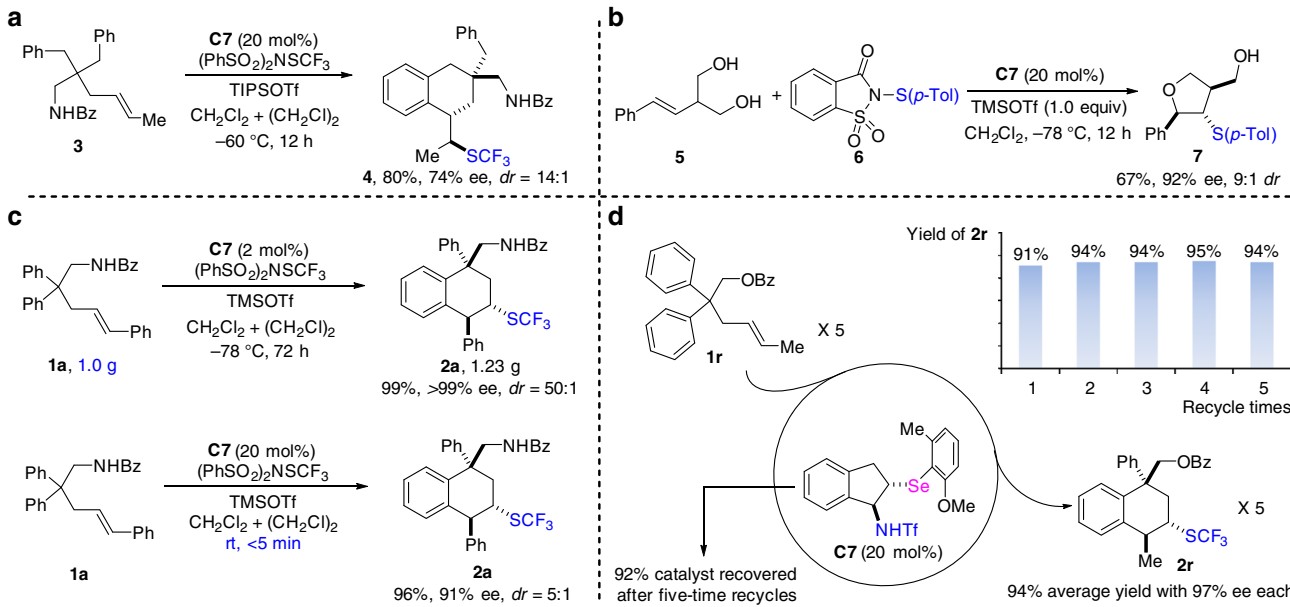

**Fig. 2** Practicability of the developed system. **a** Transformation of substrate with flexible chain. **b** Desymmetrization and sulfenylation of diols. **c** Gram-scale reaction and reaction at room temperature. **d** Recycle of the catalyst

**Fig. 3** Further transformations of products. **a** Various transformations of **2a**. **b** Intramolecular Pd-catalyzed C–H amination of **2o**

derived compounds were isolated as single isomers. First, deprotection of benzoyl group on product **2a** gave a free amine **2ab** in 96% yield. The SCF$_3$ group could be oxidized to both SOCF$_3$ and SO$_2$CF$_3$ groups by the appropriate oxidative systems. Compounds with SO$_2$CF$_3$ group could be further converted[69–71]. The generated **2ad** easily underwent the elimination of triflic group to form alkene **2ae** with Me$_3$SiOK. This provides a new route for the synthesis of valuable tetrahydronaphthalene derivatives, and shows a good potential of SCF$_3$ group in synthetic utilities. Interestingly, **2af** was formed as a diastereoisomer from **2ad** when MeONa was used as the base.

Furthermore, a spiroindoline derivative could be generated with **2o** by an intramolecular Pd-catalyzed C–H amination. In the above-mentioned transformations, the erosion of enantioselectivity was not observed.

**Computational Studies**. During the reaction for the formation of **2a**, a complex containing a chalcogenide-captured CF$_3$S cation was considered as the intermediate according to the work in which an active species was separated and could easily undergo the following step to afford the desired product for

**Fig. 4** Computational studies. Change of Gibbs free energy based on computational studies

enantioselective sulfenofunctionalization of alkenes[61]. The formation of this intermediate is the commencement of the reaction and can be affected by the used chalcogenide catalysts. On the basis of the experimental results in Table 1 and our previous studies[20], selenide catalysts are generally superior to the corresponding sulfide ones in promoting trifluoromethylthiolation, which reflects that selenides may activate CF₃S-reagent easier to generate the ion pair intermediate than sulfides. To figure out the difference between sulfide and selenide catalysts, the impact of different catalysts on the formation of chalcogenide-captured CF₃S cation was investigated. Five models with different binding interactions were proposed and the change of Gibbs free energy reflecting the difference between selenide and sulfide catalysts was calculated (Fig. 4). The results of $\Delta G$ clearly showed the huge difference caused by different catalysts. With the aid of the additive acid, the free energy for the activation of CF₃S reagent by selenide is +0.6 kcal/mol in an exothermic process, but +9.9 kcal/mol is needed to promote such step using sulfide catalyst (Fig. 4i and 4ii). When TfO⁻ anion binds to the acidic proton of the catalyst, the energy for the formation of cationic complex is largely lowered (Fig. 4ii vs. 4iii). Furthermore, when the optimal catalyst **C7** is utilized, the activation energy of the step is lowest when the methyl and methoxy groups are at the appropriate positions (Fig. 4v). These computational results match experimental ones, and indicate that high-energy barrier is required for sulfide catalysis in the initial activation step and selenide is better than sulfide in the activation of the electrophilic reagent.

**Proposed Mechanism.** On the basis of the above results and DFT calculations, a possible reaction pathway is proposed (Fig. 5a). First, selenide catalyst activates CF₃S reagent in the presence of Lewis acid to form intermediate **int-I**. Then, it reacts with substrate **1a** to afford iranium ion **int-III** through transition state **TS-I**, after which the phenyl ring on the chain attacks the iranium ion to form the final product **2a**. The reaction is spontaneous and exothermic according to calculating energies, which reasonably explains why the reaction is highly efficient under the optimal conditions. Considering the role of TfO⁻ anion in the formation of **int-I**, an anion-binding interaction with a catalyst is proposed through the entire pathway. For substrate **1a** with an NHBz group, an additional interaction between TfO⁻ and the NHBz group is suggested to construct an anion bridge in the transformation based on DFT calculations. Interestingly, the proposed anion bridge can lower the energy of the intermediates. For example, when **int-I** directly binds to substrate **1a** by hydrogen bonding, the formed intermediate has a higher energy of 2.3 kcal/mol in comparison to **int-II** (for details, see Supplementary Fig. 179). Moreover, it is noteworthy that the anion-binding interaction with the catalyst may provide a good chance for acids to participate in the construction of the chiral environment of

reaction. Especially, the effect may be more evident when the substrates without H-bonding donor groups are utilized. Because of the anion-binding interaction with catalyst, the spatial hindrance of catalytic system is modified to further fix the absolute configuration of transition states. This can be the reason why products, e.g., **2q**, **2r**, and **2s**, without H-bonding donor groups are generated in high enantioselectivities.

When calculating the reaction pathway of **1a**, it was found that the the highest energy appeared in different transition states for its four diastereomers. The highest energy is required for the attack of the phenyl ring toward the iranium ion to generate diastereomers (1R, 3S, 4S)-**2a** and (1R, 3R, 4R)-**2a**. For the formation of the other two diastereomers, the highest energy barrier lies in the step of the iranium ion formation (Fig. 5a). On the basis of the Curtin-Hammett Principle[72], the formation of **TS-I** and **TS-II** involves in the enantiodetermination of chiral centers. The energy for the formation of their possible transition states is compared (Fig. 5b). A relative $\Delta\Delta G$ (5.2 kcal/mol) for **TS-I-SRR** is obtained to predict the enantioselectivity of the major product. The predicted value is 99.9%, which is close to the experimental result (**2a**, > 99% ee). The energy discrepancy in transition states mainly comes from the perturbance of interaction and the distortion of catalyst and substrate (see the distortion-interaction analysis in Supplementary Table 4). Such two factors affect the energy of **TS-II-RRR** ($\Delta\Delta G = 1.4$ kcal/mol) and **TS-I-SSS** ($\Delta\Delta G = 3.3$ kcal/mol) as well, which result in different diastereomers of reaction ($dr_{predicted} = 37:1$). Furthermore, DFT calculations for the formation of **2q** without H-bonding interaction between substrate and TfO⁻ anion was also conducted based on the similar model. Similar results were obtained. By comparing the energy difference of the corresponding two transition states, **TS-II'-RSS** and **TS-I'-SRR**, the predicted enantioselectivity for the final product is 99.6% ee which is a little higher than the experimental result of 96% ee ($\Delta\Delta G = 3.6$ kcal/mol, see Supplementary Fig. 176 for details).

**Discussion**

In summary, we have developed an efficient approach for enantioselective desymmetrization and carbotrifluoromethylthiolation of *gem*-diaryl-tethered alkenes and alkynes to form chiral trifluoromethylthiolated tetrahydronaphthalenes by a bifunctional selenide catalyst. The desired products were obtained with excellent enantio- and diastereoselectivities. They could be further converted under mild conditions, which provided new pathways for the synthesis of various valuable tetrahydronaphthalene derivatives. The developed reaction could be scaled up to gram-scale and the catalytic system could also be used to the sulfenylation and desymmetrization of diols. These facts indicate that this method has great synthetic utility and practicality. Computational studies revealed the reason why selenide catalysis is more

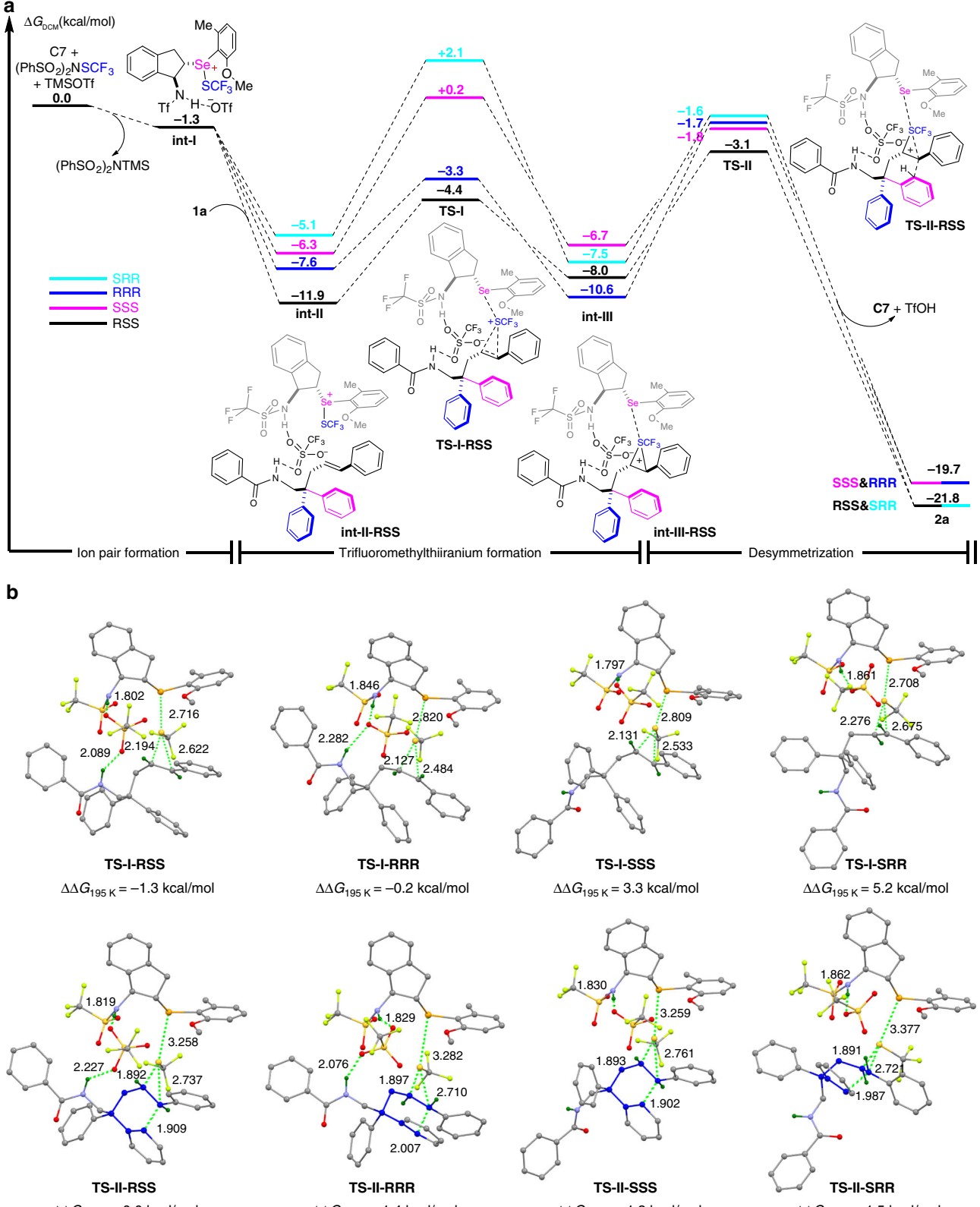

**Fig. 5** Proposed mechanism. **a** DFT calculations for reaction pathway at 195.15 K. **b** Calculated transition states related to **TS-I** and **TS-II**

efficient than sulfide catalysis, and suggested an anion-binding interaction in the whole pathway. This work constitutes an additional strategy for the synthesis of chiral tri-fluoromethylthiolated molecules, highlights the efficiency of selenide catalysis, and is complementary to Lewis base catalysis.

## Methods

**Chiral Selenide-Catalyzed Desymmetrization**. To a solution of olefin (0.1 mmol), (PhSO$_2$)$_2$N-SCF$_3$ (59.8 mg, 0.15 mmol), and catalyst **C7** (9.3 mg, 20 mol%) in solvent (CH$_2$Cl$_2$ 2 ml, (CH$_2$Cl)$_2$ 2 ml) at −78 °C was added TMSOTf (18.0 μl, 0.1 mmol). The resultant mixture was stirred at −78 °C for 12 h, and then quenched with MeOH (0.2 ml) and Et$_3$N (0.2 ml), and concentrated in vacuo. The residue

was purified by flash silica gel column chromatography to yield the corresponding CF₃S product.

**Chiral Selenide-Catalyzed Sulfenocyclization**. To a solution of olefin **5** (17.8 mg, 0.1 mmol), saccharin-S(*p*-Tol) (36.6 mg, 0.12 mmol) and catalyst **C7** (9.3 mg, 20 mol%) in solvent (CH₂Cl₂ 4 ml) at −78 °C was added TMSOTf (18.0 μl, 0.1 mmol). The resultant mixture was stirred at −78 °C for 12 h, and then quenched by saturated NaHCO₃ (1 ml) and then extracted with dichloromethane (8 ml ×4). The combined organic phases were concentrated in vacuo. The residue was purified by flash silica gel column chromatography to yield the corresponding thioproduct **7** (67%, 92% ee, 9:1 *dr*).

For nuclear magnetic resonance and high-performance liquid chromatography spectra, see Supplementary Figs 7–169.

**Data Availability**. The X-ray crystallographic coordinates for structures reported in this article have been deposited at the Cambridge Crystallographic Data Centre (CCDC), under deposition numbers CCDC 1523336, 1577179, 1532614, 1533403, and 1540104. The data can be obtained free of charge from The Cambridge Crystallographic Data Centre via http://www.ccdc.cam.ac.uk/data_request/cif. Any further relevant data are available from the authors upon reasonable request.

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

## Acknowledgements

We thank Sun Yat-Sen University, the "One Thousand Youth Talents" Program of China and the Natural Science Foundation of Guangdong Province (Grant No. 2014A030312018) for financial support. We are grateful to our teammate, Dr. Jinji Wu, for single crystal structure analysis. We thank National Supercomputing Center in Shenzhen for providing computer service for our computational studies. We also thank Professor Vy Dong at UCI for the great suggestions about the manuscript.

## Author contributions

J.L. started and performed the experiments and prepared Supplementary Information. Q. C. performed additional experiments with respect to substrate scope. X.C. performed the computational studies and revised the paper. X.Z. conceived and directed the project and wrote the manuscript.

## Additional information

**Competing interests:** The authors declare no competing financial interests.

