## [Peer Review File · Nature Communications]

Reviewers' comments:

Reviewer #1 (Remarks to the Author):

Zhao and coworkers have developed an efficient approach for enantioselective desymmetrization and carbotrifluoromethylthiolation of gem-diaryl tethered alkenes and alkynes to form chiral trifluoromethylthiolated tetrahydronaphthalenes by a bifunctional selenide catalyst. The desired products were obtained with excellent enantio- and diastereoselectivities. The developed reaction could also be scaled up to gram-scale. Despite that synthetic efficiency of the overall transformation, several issues make this chemistry not to meet the requirements for publication in a general chemistry journal like Nat. Commun. On one hand, the enantioselective trifluoromethylthiolation has already been reported by several groups (the authors cited but with no comments on these reactions). On the other hand, the author has also reported a similar enantioselective trifluoromethylthiolation using bifunctional selenide catalyst (ref.22) and the results in this manuscript are not so exciting. For these reasons, I cannot recommend publication of this manuscript in Nat. Commun. I think that a journal strictly focused on synthetic organic chemistry like Org. Lett. or Adv. Synth. Catal. would be much more appropriate for this paper.

Reviewer #2 (Remarks to the Author):

See attachment.

Reviewer #3 (Remarks to the Author):

The authors describe enantioselective trifluoromethylthiolation by desymmetrization approach. Development of the new strategy using Zhao's chiral selenide catalyst for efficient and easy access of valuable trifluoromethylthiolated tetrahydronaphthalene derivatives has the potential impact to warrant publication in Nature Communications. However, there exists insufficient experimental and computational results. The author should put more detailed additional data, in particular regarding the functional substituent effect and the computationally proposed TS model. To publish in Nature Communications, the authors should revise their manuscript according to the following suggested improvements.

(1) The authors insisted that the OTf⁻ anion bridge between catalyst and substrate play a key role in acceleration of the reaction as well as formation of the chiral space. Whereas the free OH group having coordination ability to OTf⁻ anion on substrate 2p decreased the enantioselectivity, loss of the hydrogen bond donor on substrates 2q, 2r, and 2s maintained high % ee (Table 2). What is "OH-promoted mismatched interaction"? Why do substrates

having no hydrogen bond donor group achieve high enantioselectivities? These functional substituent effects should be clarified maybe by the additional DFT calculations of the corresponding TS models. In addition, the substrate bearing Me-protected benzamide group NMeBz should be examined, giving us the straightforward evidence for fundamental role of NHBz.

(2) The substituent effect of catalyst has a great impact on the diastereoselectivity (C1 vs. C7 in Table 1). The OTf⁻ anion bridge interaction between catalyst and substrate stabilizes TS-II-RSS affording the major enantiomer in the proposed TS model (Figure 5 and distortion/interaction analysis in SI). However, the aryl group on selenium atom would not affect coordination modes with OTf⁻ anion. The additional DFT calculations of the corresponding TS models would give deeper insights in the present sophisticated catalyst design.

(3) Although the authors argued the role of acid, there exists no explanation regarding the different Lewis acids need to promote the reaction of 2p and 2y. Those experimental results should be compared with usual reaction condition using TMSOTf. Furthermore, it had better to investigate the difference between TMSOTf and TfOH because TfOH has been used in the author's previous reports (refs. 21, 22).

(4) I'm wondering about the practicability and generality of the reaction demonstrated in Figure 2b. I cannot understand the reason why olefinic diol was only employed for sulfenylation reaction. Why don't you demonstrate the usual condition using 2,2-diaryl olefinic benzamides?

Referee Report for Zhao et al. “Merging Desymmetrization and Trifluoromethylthiolation: Enantioselective Synthesis of Trifluoromethylthiolated Tetrahydronaphthalenes by Selenide Catalysis”

Overall Recommendation: Accept After Major Revisions

This manuscript builds off of a previous publication from these authors wherein the same Lewis basic selenide catalyst was employed in intramolecular aminotrifluoromethylthiolation (Ref. 22, *Org. Lett.* **2017**, *19*, 3434-3437). They expand the scope of this transformation to the desymmetrization of quaternary benzhydryl compounds and in one case a symmetric diol. Enantioselective preparation of these tetrahydronaphthalene scaffolds is undoubtedly of interest. However, the necessary limitation of a symmetric precursor providing identical functionalization at the fused ring and at the ring on the quaternary stereocenter diminishes its applicability. The presence of a heteroatom (N or O) on the carbon α - to the *gem*-diaryl also appears to be required, although this limitation is not discussed. The computational results do help some to explain the selectivity of the transformation and to some extent suggest a role for the necessary α -heteroatom. Overall, the excellent yields and selectivities, as well as the demonstrated practicability of the methodology recommend this article for publication in *Nature Communications* provided the authors rectify the issues discussed below.

Specific comments

Introduction/Background

Refs. 1-3, and especially Ref. 3, are not closely tied to the compounds that are accessible by this methodology and are primarily concerned with monofluorination and to a lesser extent trifluoromethylation.

Tetrahydronaphthalenes are obviously biologically relevant, but the structures bearing the scaffold in Figure 1 cannot be readily derived from the types of structures obtained by the methodology described in this manuscript (save for β -Belladonnine). Are there other compounds of interest with more relevant structures?

Reviews by Braga on enantioselective catalysis using organoselenium compounds were not included and should be referenced (*Curr. Org. Chem.* **2006**, 921; *Synlett* **2006**, 1453).

Citation to the book series “Lewis Base Catalysis in Organic Synthesis” should be included.

Results/Discussion

Some detail should be included (or at least have earlier work referenced) as to why each of the sulfenylating reagent, catalyst, and Lewis acid were selected for initial screening.

The best catalyst (**C7**) eventually arrived at is identical to the best catalyst employed in the *Org. Lett.* 2017 article (**C28** in that manuscript). Were these simultaneous developments, or was the simpler **C1** selected for initial screens for another reason while **C7** was already known?

Lewis acid, catalyst loading, reaction temperature, and substrate concentration were varied for certain substrates in Table 2, and it was mentioned that “modified conditions were required for different substrates according to the reactivity of the double bond or the formed – iranium ion.” This is too vague – what occurs if standard conditions are employed with these substrates? What is being corrected by adjusting the reaction conditions?

It is unclear what the authors mean by a “decrease of enantioselectivity might attribute to OH-promoted mismatched interactions between substrate and catalyst” in regards to substrate 2p.

It is also worth noting that the switch from TMS-OTf to TIPS-OTf could be at least partially responsible for the observed decrease in enantioselectivity.

There is very little in the way of functionalization on the *gem*-diaryl rings, and no examples of electron withdrawing groups. Does nucleophilic capture turn off with even weakly withdrawing groups?

The computational model provided includes a hydrogen bond in the termolecular complex **int-II**, which makes evident the potential necessity of an α -heteroatom for selectivity. However, substrates **1q-s** and **1y** are incapable of engaging in this interaction and yet give similar yields and selectivities. Do the computational results for these substrates (intermediates and transition states, relative energies of diastereomeric transition states) similarly provide results that agree with experimental data?

What is meant by the sentence “The formation of this intermediate might be the key to the entire transformation and heavily relies on the used chalcogenide catalysts” (page 5)? Obviously the Lewis base-Lewis acid complex is immediately relevant to a Lewis-base catalyzed process. Is something beyond the obvious intended by this claim?

The computational results provided required intentional inclusion of the triflate anion in the structure of **int-II**. Presumably the computed selectivities do not match experimental results if this anion is excluded from the structure. However, the section titled “The role of acid” discusses the necessity of this triflate ion in a chemical sense, that is, its physical role in the selectivity of the reaction. This is a dangerously circular argument and requires experimental evidence (e.g. employment of alternative non-coordinating anions) to corroborate. It is very noteworthy that $\text{BF}_3 \cdot \text{OEt}_2$ (Supplementary Table 1, Entry 4) gave nearly the same enantioselectivity in the absence of the supposed necessary sulfonate.

Supporting Information

Some of the ^1H NMR spectra (especially of the trifluoromethylthiolated products) contain borderline-acceptable quantities of solvent and other impurities.

HPLC traces for **2d**, **2h** and **2u** do not show baseline separation. Peak shape for **4** is borderline-acceptable.

Yields are not provided in the substrate syntheses and should be included with characterization data. Purification data *for each substrate* should be provided for reproducibility.

Weights of products should be provided in addition to percent yields.

Solid products should have melting points recorded.

Other Comments

The language used in the manuscript can be vague and often cumbersome (esp. incl. run-on sentences). The manuscript periodically switches from past to present tense. There are several overlooked typos and simple grammatical errors throughout. A few examples are listed here:

- “many efforts have been devoted to their preparation for various purposes through incorporation of fluorine atoms or fluorine-containing groups such as trifluoromethyl (CF_3), trifluoromethoxy (CF_3O) and trifluoromethanesulfonyl (CF_3S) ones into the parent molecules.”
- “By using the mixed solvents of CH_2Cl_2 and $(\text{CH}_2\text{Cl})_2$, the enantioselectivity of product **2a** could improve to >99%”
- “We envisioned that when *gem*-diaryl-tethered alkenes are employed as the substrates, the aryl group on the substrate could act as a nucleophile to directly attack chiral

selenide-captured trifluoromethylthiiranium moiety to result in the formation of chiral CF₃S-tetrahydronaphthalenes despite that electrophilic reagent-involved enantioselective desymmetrization has not been developed to construct chiral tetrahydronaphthalenes.”

A native English speaker should be asked to proofread the manuscript prior to submission to ensure for intelligibility and proper tone.

Specific Recommendations for the Authors

This reviewer recommends the following: (1) limit the introduction to desymmetrization, trifluoromethylthiolation, and Lewis base catalysis, while removing the majority of the discussion concerning the biological relevance of fluorinated molecules and tetrahydronaphthalene cores. (2) Utilize the rest of the publication space to further elaborate on the changes made to the reaction conditions for particular substrates and experimental evidence to corroborate the computationally-derived mechanism; and (3) make edits based on the **Specific Comments** section.

Response to Reviewer 1's comments:

Reviewer 1 evaluated our work based on two related points: one is *“On one hand, the enantioselective trifluoromethylthiolation has already been reported by several groups (the authors cited but with no comments on these reactions.)”*; the other is *“On the other hand, the author has also reported a similar enantioselective trifluoromethylthiolation using bifunctional selenide catalyst (ref.22) and the results in this manuscript are not so exciting.”*

Thank the reviewer for time. However, we think our work is important and exciting. To avoid a misunderstanding of our work, we would like to give detailed explanation of its significance.

In this work, we emphasize *“thus, developing new methods to create versatile chiral CF₃S-molecules, especially those with an all-carbon quaternary stereocenter through a novel and enantioselective reaction mode, is highly desirable.”* We think that it is different with our former work about asymmetric intramolecular aminotrifluoromethylthiolation realized by a traditional way (ref. 22 in the original manuscript), and does not just follow up our previous studies because of the following scientific merits:

(a) Catalytic enantioselective desymmetrization is an attractive strategy for the construction of chiral all-carbon quaternary center-containing molecules. However, electrophilic reagent-involved enantioselective desymmetrization of *gem*-diaryl-tethered alkenes/alkynes has not been developed. In this work, we used the strategy of merging desymmetrization and trifluoromethylthiolation. A library of chiral trifluoromethylthiolated tetrahydronaphthalene derivatives could be achieved with *gem*-diaryl-tethered alkenes/alkynes by selenide catalysis. The desired products bearing multiple stereocenters were obtained in high yields with excellent enantio- and diastereoselectivities (10 products with 99% or >99% ees). They could be further converted into various tetrahydronaphthalene derivatives, which provided a facile route for the synthesis of useful tetrahydronaphthalenes. The developed method could be applied to enantioselective desymmetrization of olefinic diols to afford sulfenyl cyclic ethers as well.

(b) In the field of trifluoromethylthiolation, the enantioselective introduction of CF₃S group into parent molecules is rare and challenging. Our work contributes to this field. Tetrahydronaphthalene derivatives are an important class of bioactive compounds. Using our method, CF₃S group could be efficiently incorporated into this bioactive scaffold, which provides a new route for the synthesis of valuable chiral CF₃S-molecules, especially bearing an all-carbon quaternary stereocenter.

(c) In this work, we utilized the strategy of electrophilic reagent-promoted attack of aryl group towards a multiple bond. Our work has a profound impact on enantioselective desymmetrization by the same strategy with electrophilic reagents, not only electrophilic CF₃S- and ArS-reagents.

(d) With the help of computational studies, more details such as the role of acids and the reason why selenide catalyst is more efficient than sulfide catalyst for transformations are understood. Based on the calculation results, an anion binding interaction is suggested to bind to catalyst.

(e) Furthermore, in our work, the selenide catalyst exhibits extremely high efficiency we have never had before, which will inspire researchers to conduct more studies about chiral selenide catalysis in challenging transformations.

It is true that the several examples of enantioselective trifluoromethylthiolation have been reported by different groups. But most of them focused on the enantioselective trifluoromethylthiolation of nucleophilic β -ketoesters by using cinchona alkaloids or chiral copper complexes as the catalysts. Recently, we have brought chalcogenide catalysis into the field of trifluoromethylthiolation and have developed indane-based chiral chalcogenide catalysts for the enantioselective CF_3S -lactonization and -aminocyclization. We have cited all the relevant references in the appropriate place (please see refs. 13-20 in the revised manuscript, refs. 15-22 in the original manuscript). We did not give detailed description about those works based on two reasons: (i) The detailed state of art about asymmetric trifluoromethylthiolation has been demonstrated in the introductions of a few recent papers (e.g. refs. 18 and 19). When we wrote our manuscript and tried to write the similar state of art, we felt it was wordy and unnecessary. We would like to save space to describe other more important points. So, we wrote "However, little success has been achieved on enantioselective trifluoromethylthiolation until now although stereogenic CF_3S -molecules warrant further studies considering the importance of chiral centers in medicine.¹³⁻²⁰" as you can see in the introduction of our manuscript. (ii) For this work, the key point is that we utilized the strategy of electrophilic reagent-promoted desymmetrization to construct chiral trifluoromethylthiolated molecules, not only trifluoromethylthiolation. Importantly, electrophilic reagent-involved enantioselective desymmetrization of *gem*-diaryl-tethered alkenes/alkynes has not been developed. We would like to use more sentences to introduce the desymmetrization background.

Overall, we think that this work is highly important and contributes to the fields of desymmetrization, trifluoromethylthiolation, Lewis base catalysis and tetrahydronaphthalene synthesis.

Response to Reviewer 2's comments:

Reviewer 2 commented: "*Overall Recommendation: Accept After Major Revisions. This manuscript builds off of a previous publication from these authors wherein the same Lewis basic selenide catalyst was employed in intramolecular aminotrifluoromethylthiolation (Ref. 22, Org. Lett. 2017, 19, 3434-3437). They expand the scope of this transformation to the desymmetrization of quaternary benzhydryl compounds and in one case a symmetric diol. Enantioselective preparation of these tetrahydronaphthalene scaffolds is undoubtedly of interest. However, the necessary*

limitation of a symmetric precursor providing identical functionalization at the fused ring and at the ring on the quaternary stereocenter diminishes its applicability. The presence of a heteroatom (N or O) on the carbon α - to the gem-diaryl also appears to be required, although this limitation is not discussed. The computational results do help some to explain the selectivity of the transformation and to some extent suggest a role for the necessary α -heteroatom. Overall, the excellent yields and selectivities, as well as the demonstrated practicability of the methodology recommend this article for publication in Nature Communications provided the authors rectify the issues discussed below.”

Thank the reviewer very much for the overall evaluation of our work. Some revisions have been made according to the reviewer’s suggestions and questions. Actually, pure hydrocarbon substrates work under the similar conditions. We have added the result about **2t** from pure hydrocarbon alkene substrate into Table 2 in the revised manuscript. The main reason that we chose the substrates with heteroatom on the carbon α - to the *gem*-diaryl is their synthetic convenience. More computational studies have been conducted to understand the entire pathway. Please see the following relevant discussion.

Specific comments

Introduction/Background

1) Reviewer 2 pointed out *“Refs. 1-3, and especially Ref. 3, are not closely tied to the compounds that are accessible by this methodology and are primarily concerned with monofluorination and to a lesser extent trifluoromethylation.”*

Thank the reviewer for the advice. We have deleted less relevant refs. 2 and 3.

2) Reviewer 2 pointed out *“Tetrahydronaphthalenes are obviously biologically relevant, but the structures bearing the scaffold in Figure 1 cannot be readily derived from the types of structures obtained by the methodology described in this manuscript (save for β -Belladonnine). Are there other compounds of interest with more relevant structures?”*

Thank the reviewer for pointing out. We found other compounds containing tetralin scaffold. Please see below. According to the *Specific Recommendations* of the reviewer below, we deleted the listed tetrahydronaphthalene structures in Figure 1 to save more space for experimental and computational parts.

3) Reviewer 2 pointed out *“Reviews by Braga on enantioselective catalysis using organoselenium compounds were not included and should be referenced (Curr. Org. Chem. 2006, 921; Synlett 2006, 1453).”*

Thank the reviewer for the suggestion. We have cited both reference as refs. 55 and 56 in the revised manuscript.

4) Reviewer 1 pointed out *“Citation to the book series “Lewis Base Catalysis in Organic Synthesis” should be included.”*

Thanks for the advice. The book “Lewis Base Catalysis in Organic Synthesis” has been cited as ref. 54.

Results/Discussion

5) Reviewer 2 pointed out *“Some detail should be included (or at least have earlier work referenced) as to why each of the sulfenylating reagent, catalyst, and Lewis acid were selected for initial screening.”*

Thank the reviewer for the suggestion. We have added some details in the revised manuscript: “To test the desymmetrization of **1a**, highly reactive electrophilic $(\text{PhSO}_2)_2\text{NSCF}_3$ as the CF_3S source and bifunctional catalyst **C1** based on indane scaffold were utilized (Table 1). Based on our former observations,²⁰ selenide **C1** with a triflic amide group was quite efficient for the trifluoromethylthiolation with the aid of acid.” Our previous work has been referenced.

6) Reviewer 2 pointed out *“The best catalyst (C7) eventually arrived at is identical to the best catalyst employed in the Org. Lett. 2017 article (C28 in that manuscript). Were these simultaneous developments, or was the simpler C1 selected for initial screens for another reason while C7 was already known?”*

Thank the reviewer for concerns about reaction optimization. Actually, we initiated the optimization with simpler catalyst **C1** because of its easy availability. But the results, especially about the diastereoselectivity, were not satisfactory. After the optimization of different solvents, additives and reaction temperature, the diastereoselectivity could not be further improved. So, we turned our attention to the screen of catalysts. We were surprised that the best catalyst was still **C7**.

During the reaction optimization, we also tried several other substrates as shown as follows before deciding to use **1a** as the model substrate. At the beginning, we did not know that **1a** with an NHBz group gave the best result. We optimized the reaction with the following amides and different catalysts. In most cases, unsatisfactory results were obtained. Finally, it took long time to get these current results. We have added the following results into the revised Supplementary Information.

7) Reviewer 2 pointed out “Lewis acid, catalyst loading, reaction temperature, and substrate concentration were varied for certain substrates in Table 2, and it was mentioned that “modified conditions were required for different substrates according to the reactivity of the double bond or the formed $\text{-iranium ion}.$ ” This is too vague – what occurs if standard conditions are employed with these substrates? What is being corrected by adjusting the reaction conditions?”

Thank the reviewer for pointing out. The optimal conditions are suitable for most substrates to give satisfactory results. However, during the exploration of substrate scope, we found that some substrates could not be fully converted to the desired products, and some substrates gave inseparable byproducts. When reaction temperature was raised, the reactions could go to completion. For example, for the formation of **1b-1d**, the reactions could be completed at $-60\text{ }^\circ\text{C}$. For the formation of **1e** and **1h**, possible byproducts related to **R1** or **R2** as shown as follows were observed. When lowering the substrate concentration and decreasing the catalyst loading, the formation of byproducts could be suppressed. Furthermore, using different Lewis acids, the yields and enantioselectivities of reactions were slightly adjusted. To make these points more clear, we have made changes in the revised manuscript. The current description is “Moreover, modified conditions were required for some substrates to give better yields or slightly better enantioselectivities. For

example, the reactions could not go to completion under the optimal conditions for the formation to **2b-2d** most likely because the weakly electron-withdrawing aryl group on the double bond eroded its reactivity toward CF_3S^+ cation. When the reaction temperature was raised to $-60\text{ }^\circ\text{C}$, all the substrates were fully converted. Besides, low catalyst loading (10 mol%) and low concentration were appropriate for the generation of **2e** and **2h** to suppress the possible attack of the electron-rich aryl group of catalyst toward the iranium ion.” We have also added the results for the formation of **2p**, **2t** and **2z** under the standard conditions in Table 2.

8) Reviewer 2 pointed out *“It is unclear what the authors mean by a “decrease of enantioselectivity might attribute to OH-promoted mismatched interactions between substrate and catalyst” in regards to substrate 2p. It is also worth noting that the switch from TMS-OTf to TIPS-OTf could be at least partially responsible for the observed decrease in enantioselectivity.”*

Thank the reviewer for pointing out. Sorry to make the reviewer confused. Actually, we would like to express that the decreased enantioselectivity might contribute to OH-induced inappropriate H-bonding interaction between substrate and catalyst. Either stronger or weaker H-bonding interaction can affect the chiral environment of transition state, leading to the change of the enantioselectivity of reaction. The switch from TMSOTf to TIPSOTf Lewis acid slightly affected the enantioselectivity. Please see Table 2. We put the results with both Lewis acids inside the table. To make the point clearer, we also changed the sentence to “the decrease of enantioselectivity might attribute to OH-induced inappropriate H-bonding interaction between substrate and catalyst.” in the revised manuscript.

9) Reviewer 2 pointed out *“There is very little in the way of functionalization on the gem-diaryl rings, and no examples of electron withdrawing groups. Does nucleophilic capture turn off with even weakly withdrawing groups?”*

Thank the reviewer for pointing out. Actually, we have tried to prepare some substrates with electron-withdrawing groups on the *gem*-diaryl rings such as **R3-R6**. But, it failed to synthesize **R4-R6**. Compound **R3** could be accessed, and gave messy products in full conversion under the standard conditions.

10) Reviewer 2 pointed out “*The computational model provided includes a hydrogen bond in the intermolecular complex **int-II**, which makes evident the potential necessity of an α -heteroatom for selectivity. However, substrates **1q-s** and **1y** are incapable of engaging in this interaction and yet give similar yields and selectivities. Do the computational results for these substrates (intermediates and transition states, relative energies of diastereomeric transition states) similarly provide results that agree with experimental data?*”

Thank the reviewer for these concerns. Our method also works for pure hydrocarbon substrates. We have added a result of **2t** (86% ee) in the revised manuscript, which indicates α -heteroatom is not obligatory. The main reason we chose the class of substrates with an α -heteroatom is their synthetic convenience.

To figure out whether the computational results with the substrates without an H-bonding donor group, e.g., **1q**, provide results that agree with experimental data, more computational studies have been conducted. All the pathways in the formation of four diastereomers of product **2a** were calculated as shown in the following. It was found that the highest energy appeared in different transition states for its four diastereomers. Based on the Curtin-Hammett Principle, the formation of **TS-I** and **TS-II** involves in the enantiodetermination of chiral centers. The predicted ee value is 99.9% which is close to the experimental result (**2a**, >99% ee).

Furthermore, we chose **1q** without H-bonding interaction between substrate and TfO⁻ anion as a representative substrate for DFT calculation. Based on the similar model, similar calculation results were obtained as shown in the following. The predicted enantioselectivity for the final product is 99.6% ee which is a little higher than the experimental result of 96% ee. This result is acceptable.

We have added all the calculation details in the Supplementary Information, and also made changes in the Proposed Mechanism section in the revised manuscript.

11) Reviewer 2 pointed out *“What is meant by the sentence “The formation of this intermediate might be the key to the entire transformation and heavily relies on the used chalcogenide catalysts” (page 5)? Obviously the Lewis base-Lewis acid complex is immediately relevant to a Lewis-base catalyzed process. Is something beyond the obvious intended by this claim?”*

Thank the reviewer for pointing out. Possibly, our writing is not so clear. We mean that the formation of this intermediate is the first step in the catalytic cycle. If the formation of chalcogenide captured CF_3S cation is difficult, the reaction would be certainly sluggish to lead to the poor reactivity. During our optimization, we found that selenide **C1** and sulfide **C2** catalysts exhibited huge distinction in catalytic activity. We speculated that their ability to activate the SCF_3 reagent caused this distinction. So, we described it as “heavily relies on the used chalcogenide catalysts”. In the computational discussion, the difference has been demonstrated. To make the sentence more clear, we re-wrote “The formation of this intermediate is the commencement of the reaction and can be affected by the used chalcogenide catalysts.”

12) Reviewer 2 pointed out *“The computational results provided required intentional inclusion of the triflate anion in the structure of **int-II**. Presumably the computed selectivities do not match experimental results if this anion is excluded from the structure. However, the section titled “The role of acid” discusses the necessity of this triflate ion in a chemical sense, that is, its physical role in the selectivity of the reaction. This is a dangerously circular argument and requires experimental evidence (e.g. employment of alternative non-coordinating anions) to corroborate. It is very noteworthy that $\text{BF}_3 \cdot \text{OEt}_2$ (Supplementary Table 1, Entry 4) gave nearly the same enantioselectivity in the absence of the supposed necessary sulfonate.”*

Thank the reviewer very much for this point. As we show in the manuscript, our proposed mechanism mainly relies on the computational results. Actually, we did some NMR studies to try to figure out the interaction among substrate, acid and catalyst. We did observe some changes, primarily the change of proton chemical shifts of substrate and catalyst, when mixing both, three or all of them with substrate, acid, catalyst and CF_3S -reagent. It is still hard to tell what exactly the interaction looks like based on these NMR studies. However, the computational studies help us understand more about the interaction and the role of anion.

As we show in the Supplementary Table 1, product **2a** was formed with 92% ee

under the conditions using $\text{BF}_3 \cdot \text{OEt}_2$ (versus 99.2% ee with TMSOTf, 98% ee with TfOH). When $\text{BF}_3 \cdot \text{OEt}_2$ is employed as the acid, the formed $(\text{PhSO}_2)_2\text{NBF}_3^-$ is possible to interact with catalyst, and is a little different with the TfO^- anion. Presumably, this difference leads to the decrease of the enantioselectivity of reaction.

Based on the suggestions of the reviewer, we have deleted the section of “The role of acid”, and put some discussion in the computational discussion section in the revised manuscript.

Supplementary Information

13) Reviewer 2 pointed out “Some of the ^1H NMR spectra (especially of the trifluoromethylthiolated products) contain borderline-acceptable quantities of solvent and other impurities.”

Thank the reviewer for pointing out. New ^1H NMR spectra of **2a**, **2d**, **2l**, **2p** were obtained to replace the former ones.

14) Reviewer 2 pointed out “HPLC traces for **2d**, **2h** and **2u** do not show baseline separation. Peak shape for **4** is borderline-acceptable.”

Thank the reviewer for pointing out. The HPLC traces the reviewer referred to are possible from compounds **2e**, **2i**, **2v** and **2ab**, not from **2d**, **2h**, **2u** and **4**. Incorporating the advice from the reviewer, we re-purified the racemic samples related to **2e**, **2i** and **2v**, respectively, and gained their new HPLC traces to replace the old ones. In order to get nicer HPLC trace for **2ab**, we tried to use different conditions to separate the enantiomers. We could not get it possibly because the interaction between the active hydrogen from the compound and stationary phase. However, we think that the current peak shape does not affect the measurement of the enantioselectivity.

15) Reviewer 2 pointed out “Yields are not provided in the substrate syntheses and should be included with characterization data. Purification data for each substrate should be provided for reproducibility.”

Thank the reviewer for pointing out. Yields in the substrate syntheses and purification data for each substrate have been added to the revised Supplementary Information.

16) Reviewer 2 pointed out *“Weights of products should be provided in addition to percent yields.”*

Thank the reviewer for pointing out. Weights of products have been added in the revised Supplementary Information.

17) Reviewer 2 pointed out *“Solid products should have melting points recorded.”*

Thank the reviewer for pointing out. Melting points for all the solids have been added in the revised Supplementary Information.

Other Comments

18) Reviewer 2 pointed out *“The language used in the manuscript can be vague and often cumbersome (esp. incl. runon sentences). The manuscript periodically switches from past to present tense. There are several overlooked typos and simple grammatical errors throughout. A few examples are listed here:”*

Thank the reviewer for pointing out. We have checked the paper thoroughly and some mistakes have been corrected in the revised manuscript.

- *“many efforts have been devoted to their preparation for various purposes through incorporation of fluorine atoms or fluorine-containing groups such as trifluoromethyl (CF₃), trifluoromethoxy (CF₃O) and trifluoromethanesulfonyl (CF₃S) ones into the parent molecules.”*

Thank the reviewer for pointing out. We have modified the sentence to *“In recent years, many efforts have been devoted to the incorporation of fluorine atoms or fluorine-containing groups such as trifluoromethyl (CF₃), trifluoromethoxy (CF₃O) and trifluoromethanesulfonyl (CF₃S) ones into the parent molecules for various purposes.⁴⁻⁸”*

- *“By using the mixed solvents of CH₂Cl₂ and (CH₂Cl)₂, the enantioselectivity of product **2a** could improve to >99%”*

Thank the reviewer for pointing out. We have changed the sentence to *“Using the mixed solvents of CH₂Cl₂ and (CH₂Cl)₂, the enantioselectivity of product **2a** could be improved to >99% (Table 1, entry 9).”*

- *“We envisioned that when gem-diaryl-tethered alkenes are employed as the*

substrates, the aryl group on the substrate could act as a nucleophile to directly attack chiral selenide-captured trifluoromethylthiiranium moiety to result in the formation of chiral CF₃S-tetrahydronaphthalenes despite that electrophilic reagent-involved enantioselective desymmetrization has not been developed to construct chiral tetrahydronaphthalenes.”

Thank the reviewer for pointing out. We have revised the sentences to “We envisioned that when *gem*-diaryl-tethered alkenes were employed as the substrates, the aryl group on the substrate could act as a nucleophile to attack chiral selenide-captured trifluoromethylthiiranium moiety to directly afford chiral CF₃S-tetrahydronaphthalenes, which would represent the first example of electrophilic reagent-involved enantioselective desymmetrization to construct chiral tetrahydronaphthalenes (Figure 1b).”

Specific Recommendations for the Authors

19) *“This reviewer recommends the following: (1) limit the introduction to desymmetrization, trifluoromethylthiolation, and Lewis base catalysis, while removing the majority of the discussion concerning the biological relevance of fluorinated molecules and tetrahydronaphthalene cores. (2) Utilize the rest of the publication space to further elaborate on the changes made to the reaction conditions for particular substrates and experimental evidence to corroborate the computationally-derived mechanism; and (3) make edits based on the **Specific Comments** section.”*

Thank the reviewer for these constructive suggestions. In the revised manuscripts, descriptions about biological relevance of fluorinated molecules and tetrahydronaphthalene cores have been appropriately cut. Three entries about condition screening have been added into Table 1. More details about result discussion and mechanistic studies have been added as well.

Response to Reviewer 3’s comments:

“The authors describe enantioselective trifluoromethylthiolation by desymmetrization approach. Development of the new strategy using Zhao’s chiral selenide catalyst for efficient and easy access of valuable trifluoromethylthiolated tetrahydronaphthalene derivatives has the potential impact to warrant publication in Nature Communications. However, there exists insufficient experimental and computational

results. The author should put more detailed additional data, in particular regarding the functional substituent effect and the computationally proposed TS model. To publish in Nature Communications, the authors should revise their manuscript according to the following suggested improvements.”

Thank the reviewer very much for the overall evaluation of our work. We have made some revisions in the manuscript according to the suggestions and questions. The most evident change is that we improved our computational models to make the mechanistic proposal clearer to explain the selectivity of substrates with or without an NHBz group. DFT calculation for the formation of products with substrates without H-bonding donor was also conducted based on the similar model. Similar results were obtained. Besides, the result with pure hydrocarbon substrate **1t** is added into Table 2 to show the unnecessary role of functional substituent on substrate. Additional computational results have been added into the revised Supplementary Information.

1) Reviewer 3 pointed out *“The authors insisted that the $\bar{O}Tf$ anion bridge between catalyst and substrate play a key role in acceleration of the reaction as well as formation of the chiral space. Whereas the free OH group having coordination ability to OTf^- anion on substrate **2p** decreased the enantioselectivity, loss of the hydrogen bond donor on substrates **2q**, **2r**, and **2s** maintained high % ee (Table 2). What is “OH-promoted mismatched interaction”? Why do substrates having no hydrogen bond donor group achieve high enantioselectivities? These functional substituent effects should be clarified maybe by the additional DFT calculations of the corresponding TS models. In addition, the substrate bearing Me-protected benzamide group NMeBz should be examined, giving us the straightforward evidence for fundamental role of NHBz.”*

(a) Thank the reviewer for these constructive suggestions. For the role of OH group on substrate **1p**, we would like to express that the decreased enantioselectivity might contribute to OH-induced inappropriate H-bonding interaction between substrate and catalyst as we answer question 8 from reviewer 2. Either stronger or weaker H-bonding interaction can affect the chiral environment of transition state, leading to the change of the enantioselectivity of reaction. To make this point clearer, we changed the sentence to “the decrease of enantioselectivity might attribute to OH-induced inappropriate H-bonding interaction between substrate and catalyst.” in the revised manuscript.

(b) The question “Why do substrates having no hydrogen bond donor group achieve high enantioselectivities?” is similar to question 10 from reviewer 2. To figure out the reason, more computational studies have been conducted. All the pathways in the formation of four diastereomers of product **2a** were calculated as shown in the following. It was found that the highest energy appeared in different transition states for its four diastereomers. Based on the Curtin-Hammett Principle, the formation of **TS-I** and **TS-II** involves in the enantiodetermination of chiral centers. The predicted ee value is 99.9% which is close to the experimental result (**2a**, >99% ee).

We also chose **1q** without H-bonding interaction between substrate and TfO⁻ anion as a representative substrate for DFT calculation. Based on the similar model, similar calculation results were obtained as shown in the following. The predicted enantioselectivity for the final product is 99.6% ee which is a little higher than the experimental result of 96% ee. This result is acceptable.

We have added all the calculation details in the Supplementary Information, and written the relevant discussion in the Proposed Mechanism section in the revised manuscript.

(c) Substrate with NMeBz has been examined. The result was not very satisfactory. However, it cannot be attributed to the loss of the H-bonding donor on the substrate. Presumably, NMeBz group not only enlarges steric hindrance, but also reorganizes the space arrangement of the substrate in transition states. We have included the following results in the Supplementary Information.

2) Reviewer 3 pointed out “The substituent effect of catalyst has a great impact on the diastereoselectivity (C1 vs. C7 in Table 1). The OTf- anion bridge interaction between catalyst and substrate stabilizes TS-II-RSS affording the major enantiomer in the proposed TS model (Figure 5 and distortion/interaction analysis in SI). However, the aryl group on selenium atom would not affect coordination modes with OTf anion. The additional DFT calculations of the corresponding TS models

would give deeper insights in the present sophisticated catalyst design.”

Thank the reviewer very much for this great suggestion. The substituent effect of catalyst does have a great impact on the diastereoselectivity of reaction (**C1** vs **C7** in Table 1). We attempted to figure out the reason, which can help us understand more why the product is formed in high diastereoselectivity with catalyst **C7**. The result might provide useful information for us to further design new catalysts. Based on our computational studies, interaction and distortion of catalyst and substrate begins to appear in **int-II** for the formation of **2a** although it cannot be used to accurately predict the final diastereoselectivity of reaction. Due to the computation convenience, we compared model **int-II-RSS-C7** from catalyst **C7** with the model of its analogue from catalyst **C1** as shown in the following.

According to the calculation, we find that there are three different important parts between the above models, which may be responsible for the difference of diastereoselectivity:

(a) The spatial site of TfO^- anion. Using different catalysts, the spatial site of the TfO^- anion is different. In comparison to **int-II-RSS-C1**, the TfO^- anion in **int-II-RSS-C7** moves further toward the NHTf part or phenyl ring of substrate and the NHTf group of catalyst.

(b) The distance between the aryl ring with methoxy group and the phenyl ring connecting to the double bond. The distance is shorter in **int-II-RSS-C7** than in **int-II-RSS-C1**. This will increase the steric hindrance at the double bond side, and hamper possible bond rotation. These are beneficial to the increase of chiral environment.

(c) The distance between the double bond and the SCF_3 group. Interaction between the double bond and the SCF_3 group is weaker in **int-II-RSS-C7** than in **int-II-RSS-C1** because of the elongation of the distance.

The distance shown in the following:

Spatial analysis of TfO⁻ (Hide CF₃ of TfO⁻):

Shortest distance between aryl selenide and phenyl ring of alkene:

Length of SCF₃ cation to double bond:

Å	SRR	RRR	SSS	RSS
C1	3.167 3.196	3.009 3.024	3.084 3.212	2.874 3.146
C7	3.180 3.224	3.007 3.134	2.970 3.189	3.051 3.619

In addition, the calculated energies for **int-II-RSS-C7** and **int-II-RSS-C1** and distortion interaction analysis are shown as follows. As you can see, $\Delta\Delta G$ between the enantiomers, e.g. **RSS** and **SRR**, from **C7** as the catalyst is always bigger than from **C1** as the catalyst. The larger discrepancy of energy related to **RSS** implicates stronger chiral recognition ability. These results show the substituent effect of catalyst in a way.

Distortion interaction analysis at 195.15 K in kcal/mol for int-II-RSS-C7

Entry	$\Delta E_{\text{dist-cat}}$	$\Delta E_{\text{dist-sub}}$	ΔE_{dist}	ΔE_i	ΔE_{act}	$\Delta\Delta G$
Int-II-RSS	1.2	2.5	3.7	-24.0	-20.3	0.0
Int-II-SRR	3.3	0.8	4.1	-17.1	-12.9	6.9
Int-II-SSS	2.7	0.4	3.2	-17.8	-14.7	5.6
Int-II-RRR	3.1	3.2	6.3	-22.8	-16.5	4.3

$$\Delta E_{\text{dist}} = \Delta E_{\text{dist-cat}} + \Delta E_{\text{dist-sub}}; \Delta E_{\text{act}} = \Delta E_{\text{dist}} + \Delta E_i.$$

Distortion interaction analysis at 195.15 K in kcal/mol for int-II-RSS-C1

Entry	$\Delta E_{\text{dist-cat}}$	$\Delta E_{\text{dist-sub}}$	ΔE_{dist}	ΔE_i	ΔE_{act}	$\Delta\Delta G$
Int-II-RSS	2.8	2.8	5.5	-24.0	-18.4	0.0
Int-II-SRR	4.1	0.8	4.9	-16.6	-11.7	5.9
Int-II-SSS	1.4	0.4	1.8	-17.1	-15.3	2.8
Int-II-RRR	2.0	3.5	5.5	-21.0	-15.5	2.9

$$\Delta E_{\text{dist}} = \Delta E_{\text{dist-cat}} + \Delta E_{\text{dist-sub}}; \Delta E_{\text{act}} = \Delta E_{\text{dist}} + \Delta E_i.$$

3) Reviewer 3 pointed out “Although the authors argued the role of acid, there exists no explanation regarding the different Lewis acids need to promote the reaction of **2p** and **2y**. Those experimental results should be compared with usual reaction condition using TMSOTf. Furthermore, it had better to investigate the difference between TMSOTf and TfOH because TfOH has been used in the author’s previous reports (refs. 21, 22).”

Thank the reviewer for these suggestions. To make the point about the effect of different acids in reactions more clear, we have added three entries in Table 1 (please see entries 10-12), and have also added the results with usual reaction conditions using TMSOTf for products **2p**, **2t** and **2z** in Table 2 in the revised manuscript. As you can see the results (**2a**, 98% ee with TfOH vs 99% ee with TMSOTf; **2p**, 79 ee% with TMSOTf vs 81% ee with TIPSOTf; **2t**, 83 ee% with TMSOTf vs 86% ee with TIPSOTf; **2z**, 90 ee% with TMSOTf vs 94% ee with BF_3OEt_2) in the manuscript, there is no big difference using TfOH and TMSOTf. Usually, TMSOTf gives a little better results than TfOH, and sometimes TIPSOTf gives a little better results than other Lewis acids. We

do not know exactly the reason about this slightly better effect so far. Possibly, the resulting byproduct such as $(\text{PhSO}_2)_2\text{NTMS}$ or $(\text{PhSO}_2)_2\text{NH}$ exists in reaction to affect the property of solvent or slightly interact with certain group of intermediates, which leads to the small change of enantioselectivity of reaction. Moreover, when we did NMR experiments to probe the difference with the model substrate **1a** and acids (TfOH and TMSOTf), nearly same proton shifts were observed.

4) Reviewer 3 pointed out “I’m wondering about the practicability and generality of the reaction demonstrated in Figure 2b. I cannot understand the reason why olefinic diol was only employed for sulfenylation reaction. Why don’t you demonstrate the usual condition using 2,2-diaryl olefinic benzamides?”

Thank the reviewer for pointing out. Actually, we have tried desymmetrization and sulfenylation of 2,2-diaryl olefinic benzamides. Interestingly, some unexpected results were obtained using two typical substrates. Using **1a** as the substrate, no reaction took place. Using **1i** as the substrate, aminocyclization product was formed with <60% ee. [redacted] Owing to this work about desymmetrization, we just show one desymmetrization example of the formation of product **7** in Figure 2.

[redacted]

We submitted the revised manuscript, the revised Supplementary Information and the versions with yellow highlights of what changes we have made. If there are any

questions, please feel free to let me know. Thank you for your time again.

Sincerely,

Xiaodan Zhao

Plus: The original reports from three reviewers in the following.

Reviewer #1 (Remarks to the Author):

Zhao and coworkers have developed an efficient approach for enantioselective desymmetrization and carbotrifluoromethylthiolation of gem-diaryl tethered alkenes and alkynes to form chiral trifluoromethylthiolated tetrahydronaphthalenes by a bifunctional selenide catalyst. The desired products were obtained with excellent enantio- and diastereoselectivities. The developed reaction could also be scaled up to gram-scale. Despite that synthetic efficiency of the overall transformation, several issues make this chemistry not to meet the requirements for publication in a general chemistry journal like Nat. Commun. On one hand, the enantioselective trifluoromethylthiolation has already been reported by several groups (the authors cited but with no comments on these reactions). On the other hand, the author has also reported a similar enantioselective trifluoromethylthiolation using bifunctional selenide catalyst (ref.22) and the results in this manuscript are not so exciting. For these reasons, I cannot recommend publication of this manuscript in Nat. Commun. I think that a journal strictly focused on synthetic organic chemistry like Org. Lett. or Adv. Synth. Catal. would be much more appropriate for this paper.

Reviewer #2 (Remarks to the Author):

Referee Report for Zhao et al. "Merging Desymmetrization and Trifluoromethylthiolation: Enantioselective Synthesis of Trifluoromethylthiolated Tetrahydronaphthalenes by Selenide Catalysis"

Overall Recommendation: Accept After Major Revisions

This manuscript builds off of a previous publication from these authors wherein the same Lewis basic selenide catalyst was employed in intramolecular aminotrifluoromethylthiolation (Ref. 22, *Org. Lett.* **2017**, *19*, 3434-3437). They expand the scope of this transformation to the desymmetrization of quaternary benzhydryl compounds and in one case a symmetric diol. Enantioselective preparation of these tetrahydronaphthalene scaffolds is undoubtedly of interest. However, the necessary limitation of a symmetric precursor providing identical functionalization at the fused ring and at the ring on the quaternary stereocenter diminishes its applicability. The presence of a heteroatom (N or O) on the carbon α - to the gem-diaryl also appears to be required, although this limitation is not discussed. The computational results do help some to explain the selectivity of the transformation and to some extent suggest a role for the necessary α -heteroatom. Overall, the excellent yields and selectivities, as well as the demonstrated practicability of the methodology recommend this article for publication in Nature Communications provided the authors rectify the issues discussed below.

Specific comments

Introduction/Background

Refs. 1-3, and especially Ref. 3, are not closely tied to the compounds that are accessible by this methodology and are primarily concerned with monofluorination and to a lesser extent trifluoromethylation.

Tetrahydronaphthalenes are obviously biologically relevant, but the structures bearing the scaffold in Figure 1 cannot be readily derived from the types of structures obtained by the methodology described in this manuscript (save for β -Belladonnine). Are there other compounds of interest with more relevant structures?

Reviews by Braga on enantioselective catalysis using organoselenium compounds were not included and should be referenced (*Curr. Org. Chem.* **2006**, 921; *Synlett* **2006**, 1453).

Citation to the book series "Lewis Base Catalysis in Organic Synthesis" should be included.

Results/Discussion

Some detail should be included (or at least have earlier work referenced) as to why each of the sulfonylating reagent, catalyst, and Lewis acid were selected for initial screening.

The best catalyst (**C7**) eventually arrived at is identical to the best catalyst employed in the *Org. Lett.* **2017** article (**C28** in that manuscript). Were these simultaneous developments, or was the simpler **C1** selected for initial screens for another reason while **C7** was already known?

Lewis acid, catalyst loading, reaction temperature, and substrate concentration were varied for certain substrates in Table 2, and it was mentioned that "modified conditions were required for different substrates according to the reactivity of the double bond or the formed -iranium ion ." This is too vague – what occurs if standard conditions are employed with these substrates? What is being corrected by adjusting the reaction conditions?

It is unclear what the authors mean by a "decrease of enantioselectivity might attribute to OH-promoted mismatched interactions between substrate and catalyst" in regards to substrate **2p**. It is also worth noting that the switch from TMS-OTf to TIPS-OTf could be at least partially responsible for the observed decrease in enantioselectivity.

There is very little in the way of functionalization on the gem-diaryl rings, and no examples of electron withdrawing groups. Does nucleophilic capture turn off with even weakly withdrawing groups?

The computational model provided includes a hydrogen bond in the intermolecular complex **int-II**, which makes evident the potential necessity of an α -heteroatom for selectivity. However, substrates **1q-s** and **1y** are incapable of engaging in this interaction and yet give similar yields and selectivities. Do the computational results for these substrates (intermediates and transition states, relative energies of diastereomeric transition states) similarly provide results that agree with experimental data?

What is meant by the sentence "The formation of this intermediate might be the key to the entire transformation and heavily relies on the used chalcogenide catalysts" (page 5)? Obviously the Lewis base-Lewis acid complex is immediately relevant to a Lewis-base catalyzed process. Is something beyond the obvious intended by this claim?

The computational results provided required intentional inclusion of the triflate anion in the structure of **int-II**. Presumably the computed selectivities do not match experimental results if this anion is excluded from the structure. However, the section titled "The role of acid" discusses the necessity of this triflate ion in a chemical sense, that is, its physical role in the selectivity of the reaction. This is a dangerously circular argument and requires experimental evidence (e.g. employment of alternative non-coordinating anions) to corroborate. It is very noteworthy that $\text{BF}_3 \cdot \text{OEt}_2$ (Supplementary Table 1, Entry 4) gave nearly the same enantioselectivity in the absence of the supposed necessary sulfonate.

Supporting Information

Some of the ¹H NMR spectra (especially of the trifluoromethylthiolated products) contain borderline-acceptable quantities of solvent and other impurities.

HPLC traces for **2d**, **2h** and **2u** do not show baseline separation. Peak shape for **4** is borderline-acceptable.

Yields are not provided in the substrate syntheses and should be included with characterization data. Purification data for each substrate should be provided for reproducibility.

Weights of products should be provided in addition to percent yields.

Solid products should have melting points recorded.

Other Comments

The language used in the manuscript can be vague and often cumbersome (esp. incl. run-on sentences). The manuscript periodically switches from past to present tense. There are several overlooked typos and simple grammatical errors throughout. A few examples are listed here:

- “many efforts have been devoted to their preparation for various purposes through incorporation of fluorine atoms or fluorine-containing groups such as trifluoromethyl (CF₃), trifluoromethoxy (CF₃O) and trifluoromethanesulfonyl (CF₃S) ones into the parent molecules.”
- “By using the mixed solvents of CH₂Cl₂ and (CH₂Cl)₂, the enantioselectivity of product **2a** could improve to >99%”
- “We envisioned that when gem-diaryl-tethered alkenes are employed as the substrates, the aryl group on the substrate could act as a nucleophile to directly attack chiral selenide-captured trifluoromethylthiiranium moiety to result in the formation of chiral CF₃S-tetrahydronaphthalenes despite that electrophilic reagent-involved enantioselective desymmetrization has not been developed to construct chiral tetrahydronaphthalenes.”

A native English speaker should be asked to proofread the manuscript prior to submission to ensure for intelligibility and proper tone.

Specific Recommendations for the Authors

This reviewer recommends the following: (1) limit the introduction to desymmetrization, trifluoromethylthiolation, and Lewis base catalysis, while removing the majority of the discussion concerning the biological relevance of fluorinated molecules and tetrahydronaphthalene cores. (2) Utilize the rest of the publication space to further elaborate on the changes made to the reaction conditions for particular substrates and experimental evidence to corroborate the computationally-derived mechanism; and (3) make edits based on the **Specific Comments** section.

Reviewer #3 (Remarks to the Author):

The authors describe enantioselective trifluoromethylthiolation by desymmetrization approach. Development of the new strategy using Zhao's chiral selenide catalyst for efficient and easy access of valuable trifluoromethylthiolated tetrahydronaphthalene derivatives has the potential impact to warrant publication in Nature Communications. However, there exists insufficient experimental and computational results. The author

should put more detailed additional data, in particular regarding the functional substituent effect and the computationally proposed TS model. To publish in Nature Communications, the authors should revise their manuscript according to the following suggested improvements.

(1) The authors insisted that the OTf⁻ anion bridge between catalyst and substrate play a key role in acceleration of the reaction as well as formation of the chiral space. Whereas the free OH group having coordination ability to OTf⁻ anion on substrate 2p decreased the enantioselectivity, loss of the hydrogen bond donor on substrates 2q, 2r, and 2s maintained high % ee (Table 2). What is "OH-promoted mismatched interaction"? Why do substrates having no hydrogen bond donor group achieve high enantioselectivities? These functional substituent effects should be clarified maybe by the additional DFT calculations of the corresponding TS models. In addition, the substrate bearing Me-protected benzamide group NMeBz should be examined, giving us the straightforward evidence for fundamental role of NHBz.

(2) The substituent effect of catalyst has a great impact on the diastereoselectivity (C1 vs. C7 in Table 1). The OTf⁻ anion bridge interaction between catalyst and substrate stabilizes TS-II-RSS affording the major enantiomer in the proposed TS model (Figure 5 and distortion/interaction analysis in SI). However, the aryl group on selenium atom would not affect coordination modes with OTf⁻ anion. The additional DFT calculations of the corresponding TS models would give deeper insights in the present sophisticated catalyst design.

(3) Although the authors argued the role of acid, there exists no explanation regarding the different Lewis acids need to promote the reaction of 2p and 2y. Those experimental results should be compared with usual reaction condition using TMSOTf. Furthermore, it had better to investigate the difference between TMSOTf and TfOH because TfOH has been used in the author's previous reports (refs. 21, 22).

(4) I'm wondering about the practicability and generality of the reaction demonstrated in Figure 2b. I cannot understand the reason why olefinic diol was only employed for sulfenylation reaction. Why don't you demonstrate the usual condition using 2,2-diaryl olefinic benzamides?

REVIEWERS' COMMENTS:

Reviewer #2 (Remarks to the Author):

In this revised manuscript, the authors have satisfactorily address all of the concerns of this reviewer. The manuscript is now suitable for publication with one proviso. The enantioselective carbosulfonylation of isolated alkenes to form enantioenriched decalins under catalysis by chiral selenium containing catalysts (exactly what is reported here) has been demonstrated by Denmark. The authors need to explicitly point this out in the text and provide the following citations. Burying this information in a cluster of references is disingenuous.

J. Am. Chem. Soc. 2013, 135, 6419-6422.

J. Am. Chem. Soc. 2014, 136, 3655-3663

J. Org. Chem. 2014, 79, 140-171.

Reviewer #3 (Remarks to the Author):

The manuscript has been improved properly. Additional computational studies make the proposed mechanism clearer to explain the stereochemical outcome. For the benefit of readers, the author should further add some experimental results regarding to the substituent effect (response 1c) and the substrate limitation for sulfonylation (response 4) in the main text. This paper is now acceptable for publication in Nature Comm. after the minor corrections described above.

Response to Reviewer 2's comments:

Reviewer 2 pointed out: "In this revised manuscript, the authors have satisfactorily address all of the concerns of this reviewer. The manuscript is now suitable for publication with one proviso. The enantioselective carbosulfonylation of isolated alkenes to form enantioenriched decalins under catalysis by chiral selenium containing catalysts (exactly what is reported here) has been demonstrated by Denmark. The authors need to explicitly point this out in the text and provide the following citations. Burying this information in a cluster of references is disingenuous. J. Am. Chem. Soc. 2013, 135, 6419-6422. J. Am. Chem. Soc. 2014, 136, 3655-3663. J. Org. Chem. 2014, 79, 140-171."

Thank the reviewer very much for the positive evaluation of our work. Taking the advice, we have pointed out the point in the main text and cited the three references in the appropriate place. On page 4, we have added "When this method was applied to the desymmetrization and sulfonylation of **1a** with sulfonylating reagents, no reaction occurred. This result was unexpected since the carbosulfonylation of alkenes

has been realized by chiral selenophosphoramidate catalysis.⁶⁶⁻⁶⁸

Response to Reviewer 3's comments:

Reviewer 3 pointed out "The manuscript has been improved properly. Additional computational studies make the proposed mechanism clearer to explain the stereochemical outcome. For the benefit of readers, the author should further add some experimental results regarding to the substituent effect (response 1c) and the substrate limitation for sulfenylation (response 4) in the main text. This paper is now acceptable for publication in Nature Comm. after the minor corrections described above."

Thank the reviewer very much for the approval of our revision. Taking the advice, we have added the description of experimental results regarding the substituent effect in the main text. On page 2, we have added "It is noteworthy that the reaction could not go to completion and the corresponding product was formed in moderate selectivity under the optimal conditions when the substrate derived from **1a** by further protecting nitrogen with methyl group was used (63% ee, see Supplementary Table 3 for details)." We have added the result from **1o'** (Me group-protected sulfonamide) into Table 2 as well. The relevant description "In contrast, when the nitrogen of **1o** was protected by methyl group, the corresponding substrate **1o'** gave product **2o'** with lower enantioselectivity (85% ee)." has been added in the main text.

Regarding the substrate limitation for sulfenylation, we have added the description in the main text as we respond to the suggestion from reviewer 2.